# A Feasibility Study towards the On-Line Quality Assessment of Pesto Sauce Production by NIR and Chemometrics

**DOI:** 10.3390/foods12081679

**Published:** 2023-04-18

**Authors:** Daniele Tanzilli, Alessandro D’Alessandro, Samuele Tamelli, Caterina Durante, Marina Cocchi, Lorenzo Strani

**Affiliations:** 1Department of Chemical and Geological Sciences, University of Modena and Reggio Emilia, Via Campi 103, 41125 Modena, Italy; daniele.tanzilli@unimore.it (D.T.); alessandro.dalessandro@barilla.com (A.D.); samueletamelli1997@gmail.com (S.T.); caterina.durante@unimore.it (C.D.); lostrani@unimore.it (L.S.); 2Université de Lille, CNRS, LASIRE (UMR 8516), Laboratoire Avancé de Spectroscopie pour les Interactions, la Réactivité et l’Environnement, 59000 Lille, France

**Keywords:** MSPC charts, on-line, process monitoring, NIR, Basil, pesto production, PCA, PLS

## Abstract

The food industry needs tools to improve the efficiency of their production processes by minimizing waste, detecting timely potential process issues, as well as reducing the efforts and workforce devoted to laboratory analysis while, at the same time, maintaining high-quality standards of products. This can be achieved by developing on-line monitoring systems and models. The present work presents a feasibility study toward establishing the on-line monitoring of a pesto sauce production process by means of NIR spectroscopy and chemometric tools. The spectra of an intermediate product were acquired on-line and continuously by a NIR probe installed directly on the process line. Principal Component Analysis (PCA) was used both to perform an exploratory data analysis and to build Multivariate Statistical Process Control (MSPC) charts. Moreover, Partial Least Squares (PLS) regression was employed to compute real time prediction models for two different pesto quality parameters, namely, consistency and total lipids content. PCA highlighted some differences related to the origin of basil plants, the main pesto ingredient, such as plant age and supplier. MSPC charts were able to detect production stops/restarts. Finally, it was possible to obtain a rough estimation of the quality of some properties in the early production stage through PLS.

## 1. Introduction

One of the most important aspects in food industrial production, in addition to basic safety and compliance requirements, is the capability to guarantee a constant quality of the final product, including all aspects from composition to appearance and taste. To achieve this aim, a lot of effort is spent monitoring the process, usually through univariate control charts and focusing most of the effort on monitoring the quality of the final product. However, operating in this way is not optimal when the food processing is complex, and production is massive. In fact, it may be difficult in this way to understand which are the Normal Operative Conditions (NOC) of the process. Since many parameters can change simultaneously and can be correlated, it is not easy for plant operators to detect the problem in a fast way in case anomalies or deviations occur [1]. In addition, although reference analyses are reliable and efficient in assessing the final product’s quality, they provide slow responses as the sample must be collected, brought into the laboratory, and analyzed, being, at the same time, expensive in terms of money, operators’ effort, and waste. For this reason, different types of sensors that can provide timely information are becoming more and more used for on-line quality probing from raw materials to the semi-finished and final products. It has been widely demonstrated that NIR spectroscopy has a powerful potential in monitoring food production processes [2,3,4,5,6,7,8,9,10,11], due to its ability to detect both chemical and physical changes in the samples. To cite a few applications: NIR has been used for process monitoring in the dairy industry, from the prediction of raw milk composition to milk coagulation in cheese production and yogurt fermentation [11]; the fermentation processes in the wine and brewery industries; and the powdered ingredients mixing stage in different food matrices [10]. Thus, the on-line implementation of a NIR monitoring system is desired for several reasons: the timely handling of any possible faults, reducing products out of specification, thus reducing waste and economical loss. Moreover, if in addition to the data coming from process sensors controlling the machinery settings (such as the temperature, mixing rate, pressure, etc.) fused with NIR, it could become feasible to achieving a better understanding of processes, which could aid in designing more efficient and environmentally friendly processes [12,13]. However, it is still not so common in food production to have implemented systems for the data storage of retrieval process sensors. Nonetheless, companies are becoming increasingly interested in developing models that can achieve real-time monitoring and improve industrial processes.

However, it is difficult to handle, fuse, and interpret sensor data, as it is not possible to rapidly extract useful information from spectra and images without proper statistical tools. Thus, developing multivariate control charts based on latent variables and real-time prediction models, benefitting from the chemometric development in this area, is starting to be a recognized advantage in the industry.

It has been extensively demonstrated how Multivariate Statistical Process Monitoring/Control based on Latent Variables (MSPC-LVs) can lead to an efficient process monitoring [14,15,16,17,18,19,20,21,22].

The present work concerns a feasibility study to set up a model for the on-line monitoring of the pesto production process in the company Barilla, where, at the moment, a vision system (RGB camera) is monitoring the main raw material, i.e., basil, and a NIR probe installed in-line is monitoring the initial semi-finished product. The main aim of this preliminary feasibility study is the evaluation of the possible advantages that MSPC-LVs based on in-line acquired data can furnish both in terms of the possibility of estimating the quality of the finite product in real time and capturing the process evolution and the eventual departure from NOC. In this context, PCA models have been used to explore the data structure and the information they furnish. Furthermore, multivariate control charts for process monitoring based on NOC data were built. Lastly, a first attempt to obtain predictive models for the real-time prediction of main pesto quality parameters has been also carried out.

The focus has been on discussing the steps that were more critical for the models’ development. Although the results are very preliminary, some interesting indications and directions for improvement could be formulated.

## 2. Materials and Methods

### 2.1. Process Description

The analyzed data were collected from the pesto sauce line during the 2020 harvesting season in a production plant owned by the company Barilla G. e R. Fratelli S.p.A., located near Parma, Italy. In this campaign, two different varieties of basil (*Ocimum basilicum*), the main ingredient of the sauce, have been provided by five local suppliers and continuously delivered to the process line. Each basil variety was harvested four times at different plant ages: the first cut was performed at 40 days, whereas the successive cuts were each carried out every 20 days. 

At the beginning of the process line, a vision system (RGB camera) was installed that acquired images of basil plants while passing on the conveyor belt. The system was set to deliver some parameters in real time, such as the average and standard deviation values (every 15 s) of the R, G, and B channels and a rough estimation of the basil leaves’ area in the acquired image (not always available at the same time intervals); however, the raw images were not always stored. Thus, in this work, only the R, G, and B parameters could be considered. 

After this step, basil was mixed with salt and oil, forming an intermediate product, which was monitored on-line by a NIR probe. Then, all the other ingredients of the sauce were added to the intermediate product to complete the production and obtain the final product, whose quality was assessed by off-line laboratory analyses. A schematic representation of the process is reported in Figure 1. A critical issue when modeling on-line data for a continuous process is to establish the process timeline to match the sensors data acquired at different time steps with the same material; in other words, the considered variables should refer to the same sample to assemble a row of the data matrix. In this case, this step revealed particularly challenging, since the mixing of the intermediate product with the other ingredients (taking place after the NIR probe) was achieved in three distinct mixers that were emptied, transferring the crude pesto to the following processing steps sequentially, ensuring a continuous material flux. Thus, the residence time was established with the experts at the plant in order to correctly match the NIR spectra, corresponding to the intermediate material with the finished pesto at the end of the line, on which the quality parameters were acquired. 

In this study, data collected from May to August 2020 were analyzed, but not all the data recorded during this period were considered for model building, due to production pauses, instrument maintenance, and unreliable acquisitions. Finally, 459 data points were considered.

This is a second critical issue when assembling the data matrix since interruptions could be quite frequent. Since the on-line RGB continue to acquire the image of the same basil when a stop occurs at the raw material conveyor belt, an inspection of the RGB parameters’ time trends with the identification of constant values as the indication of the stopping period was used. Moreover, the activation of the pump transferring the intermediate to the NIR probe was registered and was also used as an indication of stopping periods.

Finally, a data cleaning based on anomalous RGB values of the spectra was also accomplished.

### 2.2. Reference Analysis

Consistency parameters and lipids content have been considered for the assessment of pesto sauce quality. These parameters were assessed off-line by collecting pesto samples right after their production was complete.

The Consistency of pesto is evaluated measuring the flow of a standard volume of sample (100 cm^3^) under its own weight. The flow could be related to the sample viscosity. To perform the measure, a Bostwick consistometer was used (ASTM F1080-93). This is a stainless-steel slide with a reservoir of 5 × 5 × 4 cm, a mobile gate, two adjusting screws for planarity, and a track with ruler markings. The sample, conditioned to the temperature of 20 °C, was loaded into the reservoir. Then, the gate was opened, the timer was started, and sample flowed on the track. The consistency of the pesto was evaluated, measuring the distance in centimeters flowed in 30 s. Before the measure, the dedicated adjusting screws leveled the consistometer.

The total lipids content was determined by solvent extraction on a weighed sample aliquot (5 to 10 g). The extraction was conducted with an ethyl ether in a Soxhlet apparatus for 4 h. The sample was placed in a rotary evaporator and placed in an oven at 105 °C for 2 h to remove the solvent. The fat extracted was weighed at room temperature, and its content was expressed in a percentage, divided by the initial weight of the sample.

### 2.3. On-Line Instrumentation

A Sensure prototype camera (Sensure, Bergamo, Italy) was installed above the conveyor belt right after the basil plants were supplied, acquiring RGB images every 15 s. R, G, and B values were extracted by images and treated as separate variables. 

A ProFoss spectrometer (Foss, Hillerød, Denmark) was used to collect the spectra of the basil, salt and oil mixture, namely, the intermediate product. The instrument was equipped with an optical fiber, whose probe was installed at the acquisition site on the process pipe. The spectra were acquired over the 1100–1650 nm spectral range in the transmission mode, with a nominal resolution of 0–5 nm and 64 scans per sample.

### 2.4. Data Analysis

The data analysis objectives were twofold: on the one hand, we evaluated the potentiality of establishing an on-line monitoring model (Section 2.4.3: Multivariate Control Charts; the results discussed in Section 3.2) capable of describing the natural variability inherent to the process and of capturing any eventual anomalous fluctuation, and, on the other hand, we aimed at establishing predictive models (Section 2.4.4: PLS Regression; the results discussed in Section 3.3) to evaluate the feasibility of the prediction of quality properties of the pesto sauce in real time.

However, prior to the model building, multivariate data exploration (Section 2.4.2: Principal Component Analysis; the results discussed in Section 3.1) has been a mandatory step to inspect the data structure and presence of deviating samples and to establish the time points corresponding to the normal operating conditions for the plant.

To ease readability, the applied preprocessing has been enclosed and detailed in Section 2.4.1: Preprocessing.

#### 2.4.1. Preprocessing

The applied preprocessing is listed per the type of data and modelling phase:Vision System Data

The RGB data were preprocessed with autoscaling to uniformly model the variance among the different color channels.

NIR spectra prior to PCA and MSPC

NIR spectra were pre-processed to remove effects, such as scattering, introducing variability not linked with information to be retrieved, and/or to enhance extractable information. In particular, Savitzky–Golay 2nd derivative and mean centering were applied prior to exploratory Principal Component Analysis and multivariate control charts building.

NIR spectra prior to PLS regression

Savitzky–Golay 2nd derivative and mean centering were also used as preprocessing to compute the Partial Least Squares (PLS) regression model for the lipids content.

A different preprocessing strategy was needed to obtain the PLS model for consistency. This property was not directly linked to a chemical component, as the lipids show specific absorption bands that can guide the modeling; thus, it was more difficult to model, especially considering how many registered on-line spectra were influenced by any process fluctuations. Thus, in order to remove spectral variability hindering the possibility of obtaining a satisfactory calibration model, a Dynamic Orthogonal Projection (DOP) [23] algorithm was applied, using the average spectra corresponding to the same consistency values (in the calibration set) as the source data (X_source_) and the raw calibration spectra (X_tar_) as the target. The main concept in DOP is that samples showing the same (or very close) y values should show the same spectral profile; thus, the “virtual” target spectra (X*_tar_), unaffected by the influence of uncontrolled conditions, could be estimated based on a distance or association matrix (M), calculated based on the y values of the source (y_s_) and the target (y_t_) domain. The singular value decomposition (SVD) of the difference matrix among measured and virtual target spectra was then used to determine the components (A) for orthogonalization:X*_tar_ = M*×X_source_(1)
D= X_tar_ − X*_tar_(2)
 [U_A_ S_A_ V_A_] = svd(D, A)(3)
X_source_corrected_ = X_source_ (I−V_A_V_A_^T^)(4)

In our specific case, A = 4 was used after testing using from 1 to 5. 

Once the average spectra were corrected, orthogonal projection could be directly used to predict the validation set, since the correction was embedded in the model. In this case, only mean centering (of both **X** and **y**) was applied prior to PLS. 

#### 2.4.2. Principal Component Analysis

Principal Component Analysis (PCA) is a method that by decomposition of the original data **X** into two matrices **T** and **P**, [24] according to Equation (5), allow reducing the dimensionality of the data set with a large set of variables, simplifying the exploration phase and the data visualization. PCA performs a projection of data from the original variables into new variables orthogonal to each other, the Principal Components (PCs), which are a linear combination of the original ones.
**X** = **TP**^T^ + **E**(5)

If the **X** matrix was composed of *n* rows (samples) and m columns (variables), the **T** matrix, called the scores matrix, which allowed us to understand the structure of the data, was composed by *n* rows and a number of columns equal to the number of PCs, and the loadings matrix **P** was composed by a number of rows equal to m and columns equal to the number of PCs. The loadings values corresponded to the weights by which each original variable entered the linear combination, thus defining the PCs, representing the contribution of each variable to each PC. The analysis of loadings matrix allowed us to understand the correlation structure of the variables [25]. The residual matrix **E**, which represented the unmodeled information, had the same dimension of **X,** and it was obtained by the subtraction of recalculated data from the PCA model (**TP**^T^) from **X**.

#### 2.4.3. Multivariate Control Charts

PCA was also used to build multivariate control charts for MSPC. The dataset had been split in each calibration and test set manually, considering NOC observations, subdividing each period without production stops, as follows: the first part (about 65%) consisted of temporally contiguous points in the calibration set; and the second part (about 35%) was in the test set. In this way, we mimicked the real situation of continuous monitoring where samples to be predicted came after in time for each period. Observations not in NOC, as highlighted by exploratory PCA, were all included in the test set.

To estimate the correct number of PCs, cross-validation was performed with a *venetian blind* scheme with ten splits. The MSPC charts were based on two parameters: Hotelling T^2^, which described the distance of a sample in the model space, and Q, which defined the distance of a sample from the model space. In other words, if a sample had high T^2^ values, the model was able to describe it, but the distance between the sample and the center of the model was high, i.e., it showed an extreme behavior. On the other hand, if a sample was characterized by high Q values, the model was not able to describe the sample properly, hence the correlation structure of variables was different from the other samples. To assess if a sample was extreme or anomalous, signifying a departure from normal operative conditions for both control charts, the acceptance limits had to be estimated. The T^2^ limit was obtained based on Hotelling’s T^2^ distribution, whereas the Q limit was based on χ^2^ distribution and was calculated either with Jackson and Mudholkar approximation or the Box method [26,27].

#### 2.4.4. PLS Regression 

PLS is a linear regression method that allows predicting one or more response variables (Y block) from a predictor matrix (X block), establishing a multivariate linear relationship. It operates in a low-dimensional space defined by the Latent Variables (LVs), obtained from the simultaneous decomposition of **X** and **Y**, which are oriented on directions of maximum covariance between **X** and **Y** [28]. A PCA-like decomposition of **X** and **Y** is achieved (outer relation): **X** = **T P**^T^ +**E**(6)
**Y** = **U Q**^T^ + **F**
(7)
where an inner relation links the outer relation:**U** = b***T**(8)

Hence, re-expressing this as a regression model:(9)$$\overset{`}{Y}=X\;B$$
where **T** and **U** are **X** and **Y** scores, **P** and **Q** are **X** and **Y** loadings, and **E** and **F** are the residual matrices, respectively. B holds the regression coefficients that allow the prediction of **Y** from **X** directly.

Data were partitioned into calibration (70%) and validation (30%) sets by the means of a Duplex algorithm [29]. The PLS model dimensionality, i.e., the number of PLS components, was assessed by the Root Mean Square Error in Cross-Validation (RMSECV), while the Root Mean Square Error in Prediction (RMSEP) was used to evaluate the models’ predictive capability. Residual plots were also inspected.

## 3. Results and Discussion

### 3.1. Exploratory Data Analysis

Each type of data, RGB parameters, and NIR spectra were analyzed separately to visualize and explore the data structure. PCA analysis carried out on NIR spectra (acquired for 459 time points) had highlighted the presence of a cluster of samples at the negative value of PC1 and positive value of PC2, as shown in Figure 2a, as very far and different from all the other samples. Observing the PC1 versus time plot (Figure 2b), it was evident that these samples always corresponded to restarts, where production started after a period of inactivity. In Figure 2c, the loadings line plots for PC1 and PC2 are shown as the blue and red lines, respectively, where it is possible to see the absorption bands as mainly responsible for this difference. However, to jointly interpret scores and loadings plots, a PC1 vs. PC2 loadings scatter plot was also generated (Figure 2d). In the two figures, highlighted in purple, the wavelengths that describe the separation between the NOC and anomalous samples are shown. It can be observed that the band in PC1 at 1400 nm, despite being the most intense, is not involved in the description of anomalous samples but just in extreme NOC samples with high values of PC1 scores in Figure 2a. On the other hand, the bands at 1170, 1213, 1236, and 1410 nm describe the behavior of the anomalous samples, as they fell in the separation direction, meaning that these samples had very different absorptions at these wavelengths. In detail, the bands at 1178 and 1410 nm can be ascribable to lignin, namely, the second overtone of C-H bond stretching of CH_3_, and to the first overtone of the O-H bond stretching of the ROH group, respectively. Whereas, the band at 1213 and 1236 nm are related to the first and second overtone of C-H bond stretching of oleic and linoleic acid in olive oil CH_2_ [30,31]. 

Since these samples show the outliers’ behavior, as they clearly do not represent the Normal Operative Conditions (NOCs), they were removed, and a new PCA model was built in order to obtain a better visualization of the possible differences among NOC samples. 

The first PC (79.36% of variance explained) did not show any interesting trend, thus PC2 and PC3 were inspected. In Figure 3a,b, the scores plot of PC2 vs. PC3 is reported, where samples are colored according to the different additional information available, i.e., suppliers and different cuts, respectively. The suppliers’ names have not been disclosed because of confidential agreement restrictions with the company. PC2 discriminated samples according to suppliers, as almost all samples of supplier number two had positive PC2 values, and the samples of suppliers three and four had negative PC values, suggesting that they were more similar to each other, with respect to number two. Only the samples coming from supplier five did not clearly differentiate from the others, whereas the number of samples from supplier one were too low to judge. Furthermore, PC2 and PC3 could distinguish between samples related to cut one and two (negative values of PC2 and positive values of PC3), with respect to samples related to cut three and four. The possibility to discriminate against different cuts is relevant for the company, as younger basil plants generally give a higher quality product. However, observing the two plots simultaneously, it is evident that only certain suppliers, namely, number three and four, had delivered samples characterized by low cuts. In Appendix A, the loadings plots of PC2 and PC3 are reported, respectively, which show the NIR bands responsible for these differences. Even if it is not possible to assess if suppliers or cuts influence them, the PCA resulted in a valuable tool to assess if incoming information about raw materials could be linked to the intermediate product characteristics; evidently, a more systematic planning of the next harvesting campaigns could clarify if a cut or supplier were the influential factors.

PCA analysis carried out on data collected by an RGB camera was not able to detect the anomalous behavior of the samples highlighted in Figure 1. A possible explanation is that the process needed time to return to NOCs after a stop when the production restarted, and it could happen that the NIR spectra referred to material that was probably a residue of the old process (before the restart), and thus the acquired spectra did not depict the intermediate product newly produced at the beginning. Moreover, the observation of the samples’ separation due to different cuts or suppliers was less efficient than the respective analysis performed on the NIR spectra. Thus, these differences were not linked to color variation but mostly to the basil’s “chemical” profile.

### 3.2. MSPC Charts

The most interesting results related to the MSPC charts based on PCA were obtained by using the NIR data only (inclusion of RGB parameters did not provide additional insights). The PCA model, which explains 93% of the data variance with 4 Principal components, was calculated by inserting only the samples that were considered in NOCs according to plant experts in the calibration set (294 samples), whereas the test set (165 samples) comprised both NOCs and anomalous samples. The T^2^ chart, reported in Figure 4a, describes the distance of each sample from the origin within the model space. Black circles represent the calibration samples used to build the PCA model, whereas red diamonds represent the test samples projected on the model. This chart detected five groups of samples with high T^2^ values, which, again, corresponded to the NIR spectra acquired at the different restarts of the production. No other test sample exceeded the T^2^ limit. Regarding the Q chart (Figure 4b), which describes the distance of each sample from the model space, the same samples corresponding to the restart are seen anomalous as for the T^2^ chart, meaning that the model did not properly describe these samples. The charts’ limits include few non-consecutive samples and inside of the nominal 5% of the total.

Samples were also colored according to cut, supplier, consistency, and lipids values to observe if their behavior was related to these different features, but no particular trends were detected.

Nonetheless, the results obtained show how these charts are efficient in detecting possible departure from NOC, which translate to differences in intermediate products, accelerating the identification of possible plant issues or, as in this case, the adaptation of the process while returning to NOCs after a stop period. NIR is a very sensible technique to signal any variability occurring in intermediate production samples that can be due to process resetting (actual case), process drift, or variation in the NIR instrumentation setting/performance. The interpretation of the loadings and analysis of previous production campaigns data may help in discerning the different situations. 

### 3.3. Predictive Models

An attempt to obtain predictive models, which can then be possibly used to estimate the consistency and lipids content of the final product in real time, was undertaken. Since RGB data were not able to provide reliable prediction models for both parameters, only results obtained by NIR data are presented, as summarized in Table 1. 

Before model computation, data were split by using a duplex algorithm with a 70/30% proportion in the calibration and test sets, giving 142(cal)/61(test) and 33(cal)/12(test) for consistency and lipids, respectively. Afterwards, four samples belonging to the anomalous group of observations, detected by using the T^2^ and Q distances, were removed from the test set for consistency.

The prediction model for consistency was built using 9 LVs, corresponding to the minimum RMSECV (*venetian blind*, 10 splits) value. The RMSEP value was close to RMSECV (Table 1) and corresponded to an average relative percentage error of 10% in prediction, which was considered acceptable by the company for an early (intermediate product) on-line quality estimation. The samples in the test set showed a rather high variability compared to the ones in the calibration (Figure 5a,b). Nonetheless, the residuals vs. measured values of the consistency plot (Figure 5b) highlighted that the errors on both the calibration and test samples were randomly distributed, not showing any visible trend, excluding any bias.

The prediction model of total lipids content was built using a lower number of samples than the previous model, as this parameter was assessed less frequently than consistency. In this case, 5 LVs were selected according to the minimum RMSECV (*venetian blind*, 10 splits) for the model’s construction. As shown in Figure 6a, the majority of the samples had a lipid content included in the range 46–49%, and only a few samples presented higher values. This is a quite common situation in real time production, where a consistent quality of the product is pursued. In this case, a couple of samples in the test set were predicted with a higher error but, in general, the error values comprised the 2% range, which the company considered acceptable for controlling if the product was within specification for this parameter. One of the two samples with a high lipid content in the test set was predicted accurately, whereas the other one was underestimated (Figure 6b).

In Figure 7, the Variable Influence in Projection (VIP) scores are shown [32], which highlight that the band at 1166 nm, ascribable to the olive oil’s second overtone of the CH stretching of CH_3_ [30,31], is the most influent for the prediction of total lipids content. Moreover, other bands linked to lipids in olive oil [30,31] can be found at 1422 and 1461 nm, typical of the CH stretching and deformation of CH_2_, both above the significance threshold [32].

## 4. Conclusions

This study presents a feasibility study towards the real-time monitoring of an industrial food process line (pesto production). Since historical data were not available, the obtained results referred to a single basil harvesting campaign. The modeling effort concerned both latent variables based multivariate control charts, aimed at monitoring the stability of process conditions and the eventual detecting of fluctuations exceeding the natural variability of the process as well as the quality properties’ prediction in real time. Despite the fact that the collected data were limited, the results gave interesting insights, which are summarized in the following.

### 4.1. MSPC Results

(i) the RGB parameters obtained by the vision system, albeit potentially very useful, were not increasing information retrieved from NIR. We think this is due to the limited number of features extracted by the image, which could otherwise provide a good characterization of raw material; further work is in progress in this direction (e.g., detecting the percentage areas of damaged leaves, branches, and stems by an image analysis tool);

(ii) NIR-based multivariate control charts could detect restarts after temporary production stoppages, underlining that some changes occur in the intermediate product. On one hand, this is an indication of how sensible NIR spectroscopy is to monitor any changes, and, on the other hand, a monitoring system can clearly indicate when process fluctuations return to natural process variabilities and to the constancy of the product.

### 4.2. On-Line Predictive Models

(iii) The predictive model to estimate the pesto’s consistency and total lipids content, based on the NIR spectra of the intermediate product, gave errors in the external predictions, which are considered acceptable by the company for on-line quality estimation.

(iv) It is worth noting that while building predictive models of final product quality parameters based on on-line sensors data is highly desirable, they suffer from the limited response variability (which, evidently, should be confined in the in-specific ranges). When, as in this case, it is not possible to expand the calibration range by pilot studies, the models can be, nonetheless, used as a timely rough indication of the property’s value. In this respect, more than an estimation of the quality values, they may give a preliminary check about respecting specifications Within this framework, the obtained models seem promising.

Finally, it is worth mentioning the main issues encountered, such as the lack of systematic recording of acquired on-line data, the difficulties in recovering a sound synchronization scheme, and the critical role of spectral preprocessing to cope with the many sources of variabilities intrinsic in a process framework.

## Figures and Tables

**Figure 1 foods-12-01679-f001:**
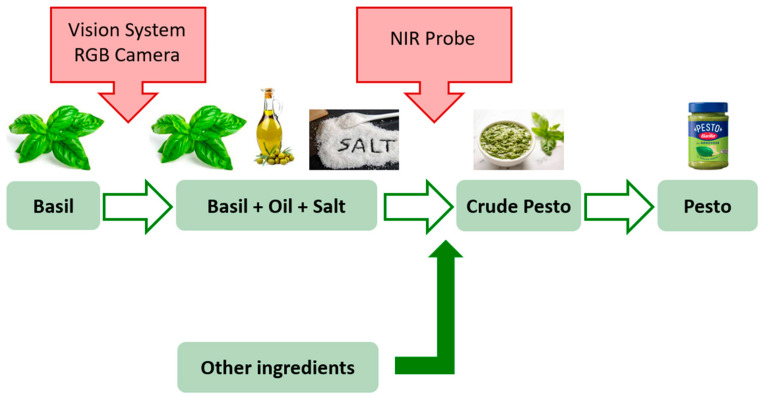
Schematic representation of pesto sauce production process.

**Figure 2 foods-12-01679-f002:**
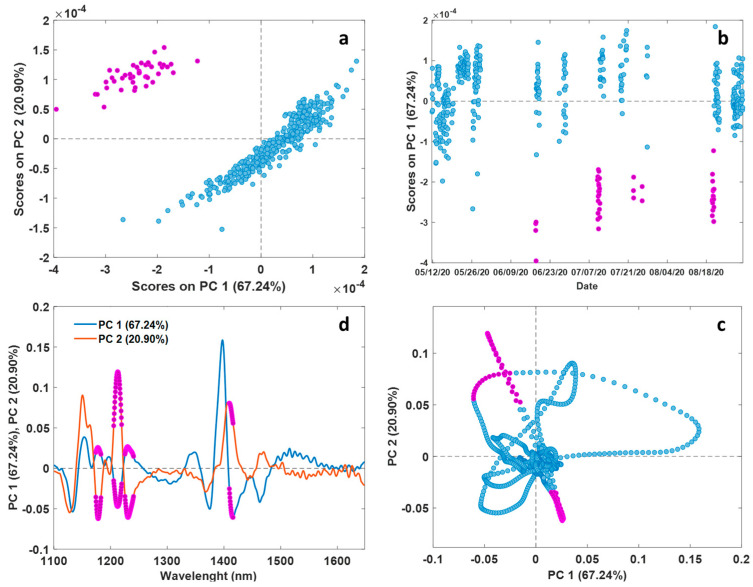
Results of the Exploratory Data Analysis performed on NIR data. PC1 vs. PC2 Scores plot (**a**), Scores on PC1 as a function of time (**b**), Loadings on PC1 and PC2 as a function of time (**c**), and Loadings on PC1 vs. PC2 (**d**). In (**a**,**b**), purple points represent anomalous samples; in (**c**,**d**), purple points represent wavelengths that mainly depict the difference of anomalous samples from the other ones.

**Figure 3 foods-12-01679-f003:**
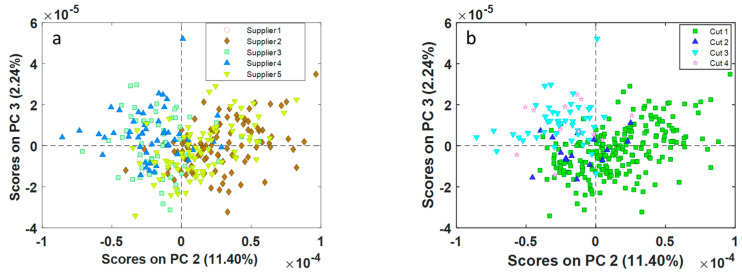
Results of the Exploratory Data Analysis performed on NIR data. PC2 vs. PC3 scores plots colored by different suppliers (**a**) and cuts (**b**).

**Figure 4 foods-12-01679-f004:**
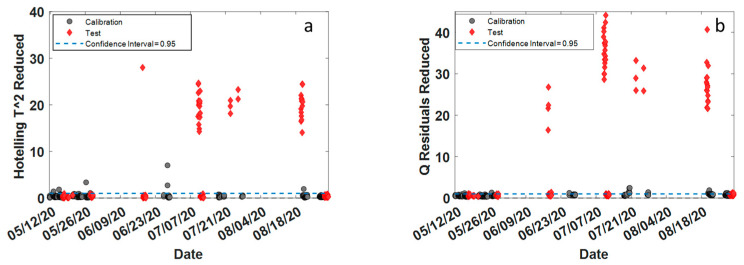
T^2^-(**a**) and Q-(**b**) based MSPC charts.

**Figure 5 foods-12-01679-f005:**
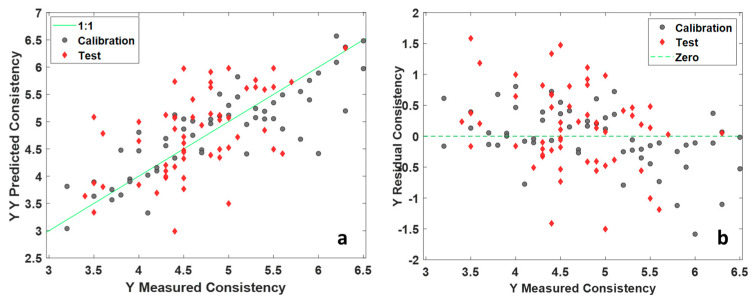
PLS results on NIR data for consistency. Predicted vs. measured values plot (**a**), residuals vs. measured values plot (**b**).

**Figure 6 foods-12-01679-f006:**
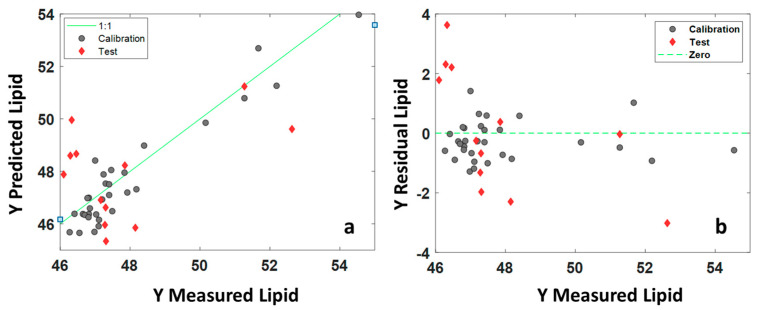
PLS results on NIR data for lipids content. Predicted vs. measured values plot (**a**), residuals vs. measured values plot (**b**).

**Figure 7 foods-12-01679-f007:**
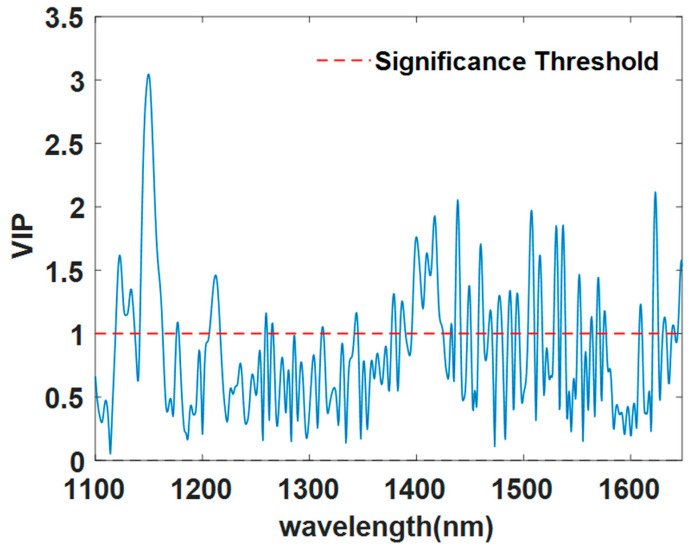
VIP scores of PLS on NIR data for lipids content.

**Table 1 foods-12-01679-t001:** Results obtained by PLS Regression.

Method	LVs	RMSECV	RMSEP
Consistency (cm)	9	0.64	0.68
Lipids (%)	5	1.59	2

## Data Availability

Data are unavailable due to privacy restrictions.

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
