# Peer review of "A Feasibility Study towards the On-Line Quality Assessment of Pesto Sauce Production by NIR and Chemometrics"

_foods, 2023, doi:10.3390/foods12081679_

Round 1

Reviewer 1 Report

In the present manuscript, A feasibility study towards on-line quality assessment of Pesto sauce production by NIR and chemometrics were evaluated.

The study is fluent and scientifically written very well. It has been tried to fill the gap that is missing from the scientific point of view. However, multivariate analyzes were included in the study. For this reason, it cannot be seen which analysis affects the results and how. For this reason, the analyzes should be simplified and explained at a level that the reader can understand.

The conclusion part is written quite long. Instead, it would be better if it was simpler and clearly emphasized what stood out.

Reviewer 2 Report

The presented researches are very interesting.

The submitted manuscript is missing some key elements.

It is not foreseen how the "quality" of the NIR analyzer measurement results is checked, in case of application of NIR probes. It's possible that this question just wasn't explained well.

It is not entirely clear why the consistency of the product and the lipid content of the product are monitored with NIR probes. Is it possible that there are some other important quality parameters that should be monitored during production?

The authors are encouraged to continue this research.

Reviewer 3 Report

1) In the introduction: application of NIR in food production with examples should be written 

2) Figure 3 could be reject.

3) PCA analysis  is often presented with cluster analysis (CA). It could be good for the paper to show also CA.

4) Why PCA was used for obtained results? Currently some other methods, such as descriptive analysis.

5) Why pesto was choose for analysis?

Round 2

Reviewer 2 Report

The authors provided good explanations for the remarks of reviewer 2.

Explanation under 1), briefly prepare and insert in additional material.

Explanation under 2), write briefly in one sentence and insert in section 2.2. Reference Analysis.

Add in the introduction a brief description of the product examined in the paper.

Also, the following improvements need to be made.

Provide the raw data used for PCA analysis and prediction models in the supplementary material.

Provide the equation and the coefficient of determination of the dependance of the predicted and measured values in Figures 5a and 6a.

Reviewer 3 Report

In my opinion, the manuscript could be published in Foods in the present form.